# DyHCN: Dynamic Hypergraph Convolutional Networks

## Abstract

Hypergraph Convolutional Network (HCN) has become a default choice for capturing high-order relations among nodes, *i.e.,* encoding the structure of a hypergraph. However, existing HCN models ignore the dynamic evolution of hypergraphs in the real-world scenarios, *i.e.,* nodes and hyperedges in a hypergraph change dynamically over time. To capture the evolution of high-order relations and facilitate relevant analytic tasks, we formulate dynamic hypergraph and devise the Dynamic Hypergraph Convolutional Networks (DyHCN). In general, DyHCN consists of a Hypergraph Convolution (HC) to encode the hypergraph structure at a time point and a Temporal Evolution module (TE) to capture the varying of the relations. The HC is delicately designed by equipping inner attention and outer attention, which adaptively aggregate nodes' features to hyperedge and estimate the importance of each hyperedge connected to the centroid node, respectively. Extensive experiments on the Tiigo and Stocktwits datasets show that DyHCN achieves superior performance over existing methods, which implies the effectiveness of capturing the property of dynamic hypergraphs by HC and TE modules.

## 1 Introduction

Graph Convolutional Network (GCN) Scarselli et al. (2008) extends deep neural networks to process graph data, which encodes the relations between nodes via propagating node features over the graph structure. GCN has become a promising solution in a wide spectral of graph analytic tasks, such as relation detection Schlichtkrull et al. (2018) and recommendation Ying et al. (2018). An emergent direction of GCN research is extending the graph covolution operations to hypergraphs, *i.e.,* hypergraph convolutional networks Zhu et al. (2017); Zhou et al. (2007); Zhang et al. (2017); Feng et al. (2019b); Yadati et al. (2019), where high-order node relations are represented as hyperedges (one hyperedge can connect multiple nodes). For instance, in a hypergraph of stocks, an financial event relevant to several stocks is represented as a hyperedge. While a surge of attention paid on hypergraph convolutional networks, most of them discard the dynamic property of hypergraphs in real-world applications, *e.g.,* new hyperedges (*i.e.,* events) emerge in the hypergraph of stocks (see Fig. 1), where the evolution of the hypergraph is crucial for the analytic tasks (*e.g.,* stock price prediction). Aiming to bridge the gap, this work explore the central theme of dynamic hypergraph and the corresponding GCN.

Formally, a hypergraph with $n$ nodes and $m$ hyperedges is represented as $\mathcal{G} = (\mathcal{V}, \mathcal{E}, \mathbf{A}, \mathbf{H}, \mathbf{X})$ where $\mathcal{V}$ and $\mathcal{E}$ denote the set of nodes and hyperedges respectively; $\mathbf{A} \in \mathbb{R}^{n \times m}$ is an incidence matrix with binary value indicating the connectedness of nodes; $\mathbf{H} \in \mathbb{R}^{m \times c}$ and $\mathbf{X} \in \mathbb{R}^{n \times d}$ are features represent the hyperedges and nodes respectively. In order to account for the evolution, we first extend the concept of static hypergraph to dynamic hypergraph, which has two different formulations when treating the time as continuous value or discrete value. 1) Discrete-time formulation. A straightforward solution is to treat a time window with length of $T$ (*e.g.,* $T$ days) as a sequence of time-steps and get a snapshot at each time-step. In this way, a dynamic hypergraph is defined as $\mathcal{G}^D = [\mathcal{G}^1, \cdots, \mathcal{G}^t, \cdots, \mathcal{G}^T]^T$ where $\mathcal{G}^t$ is a hypergraph dumped at time-step $t$. 2) Continuous formulation. By treating time as a continuous variable, the dynamic hypergraph can be defined as $\mathcal{G}^C = (\mathcal{G}^0, \mathcal{U})$ where $\mathcal{G}^0$ is the initial status (a hypergraph) and $\mathcal{U} = \{(p^t, v^t, a^t)|t <= T\}$ is a streaming of updates. $p^t$ denotes the target variable (*e.g.,* a row of $\mathbf{X}$) changed at time $t$; $v^t$ denotes the latest value of the target variable, $a^t$ denotes the action of change, including add, delete,

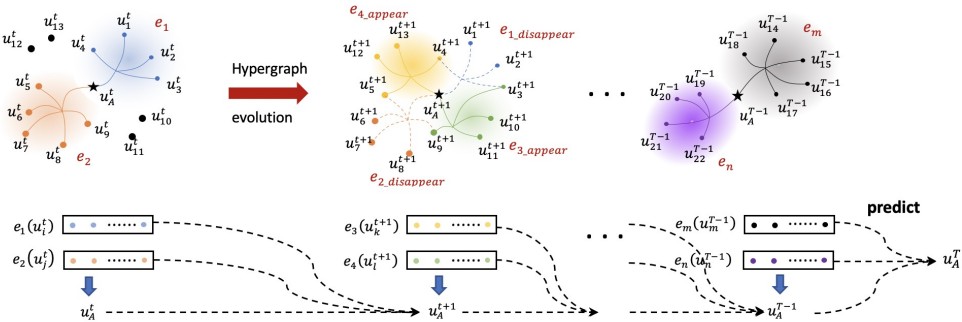

Figure 1: The evolution of dynamic hypergraph.

update. It should be noted that both formulations have pros and cons, *e.g.,* the discrete-time formulation is more friendly to existing analytic techniques on static hypergraph such as HCN while the continuous-time formulation records the accurate time of changes. This work focuses on the discrete-time formulation and makes the first attempt to extend HCN to dynamic hypergraph.

A big challenge of capturing spatial-temporal dependency in a dynamic hypergraph is that it is tough to extract the features of those changing nodes or hyperedges in a unified manner for the sake of varied scales of nodes and hyperedges. Besides, how to absorb their dynamic properties is very important for various application tasks. Towards this end, we need to design the proper convolution operations on dynamic hypergraph. There are two challenging toughs: 1) at each time step, since there are various relations between hyperedges and nodes, it is important to update the node features by considering various relations in the hyperedges; 2) due to dynamically changes of the node features, modeling the temporal dependency needs to extract the corresponding temporal features.

In this work, we propose a framework of Dynamic Hypergraph Convolutional Networks (DyHCN) to tackle the challenges, which has two modules: Hypergraph Convolution (HC) module and Temporal Evolution (TE) module. In a dynamic hypergraph, the set of hyperedges at each time step includes different hyperedge embeddings and each hyperedge contains different numbers of nodes. We exploit three submodules to update an node's embeddings in HC: inner attention, outer attention, and embeddings update. Firstly, inner attention transform node features along with its hyperedge into the node-hyperedge feature; and then outer attention utilizes attention mechanism to estimate the importance of each hyperedge and output the importance weights; and then we update the node's embeddings by aggregating node-hyperedge, hyperedge and node features with the weight of each hyperedge. Getting the nodes embeddings, we extract temporal features of nodes' embeddings and make a prediction by the TE module. Extensive experimental results on two real-world datasets validate the superior performance of DyHCN over the existing baselines which proves the effectiveness of DyHCN on dynamically hypergraphs.

The rest of the paper is organized as follows. Section 2 introduces the preliminary knowledge about GCN and the hypergraph convolutional network. Section 3 explains the proposed DyHCN method. Section 4 introduces related work about GCN on the graph and hyperedge. Applications and experimental results are presented in Section 5. Finally, we conclude this work in Section 6.

## 2 PRELIMINARY

**Graph Convolutional Network**    Given a graph $\mathcal{G} = (\mathcal{V}, \mathcal{E})$ with $N$ nodes $v_i \in \mathcal{V}$, edges $(v_i, v_j) \in \mathcal{E}$, an adjacency matrix $\mathbf{A} \in \mathbb{R}^{N \times N}$ and a degree matrix $\mathbf{D}_{ii} = \sum_j \mathbf{A}_{ij}$. With the input signal $\mathbf{x}$, Kipf & Welling (2016) considers spectral convolutions on graphs with a filter $g_\theta = diag(\theta)$ in the Fourier domain, $g_\theta \star \mathbf{x} = \mathbf{U} g_\theta \mathbf{U}^T \mathbf{x}$, where $\mathbf{U}$ is the matrix of eigenvectors of the normalized graph Laplacian $\mathbf{L} = \mathbf{I}_N - \mathbf{D}^{-1/2} \mathbf{A} \mathbf{D}^{-1/2} = \mathbf{U} \Lambda \mathbf{U}^T$, with a diagonal matrix of eigenvalues $\Lambda$ and the graph Fourier transform $\mathbf{U}^T x$. In order to reduce the computation complexity, $g_\theta$ is approximated with Chebyshev polynomials $T_k(x) = 2x T_{k-1}(x) - T_{k-2}(x)$ Defferrard et al. (2016), which can be formulated as: $g_\theta \approx \sum_{k=0}^{K} \theta_k T_k(\hat{\Lambda})$, where $\hat{\Lambda} = \frac{2}{\lambda_{max}} \Lambda - \mathbf{I}$, $\lambda_{max}$ denotes the largest eigenvalue

of Laplacian matrix, $\theta_k$ denotes the Chebyshev coefficients. Kipf & Welling (2016) proved that the GCN can be simplified to $K=1$ and $\lambda_{max} \approx 2$, which is the state-of-the-art of GCN.

**Hypergraph Convolutional Network**  A hypergraph can be formulated as $\mathcal{G} = (\mathcal{V}, \mathcal{E}, \mathbf{W})$, where $\mathcal{V}$ is a set of vertes, $\mathcal{E}$ is a set of hyperedges and $\mathbf{W}$ is a diagonal matrix which denotes the weight of each hyperedge. The adjacency matrix of hypergraph $\mathcal{G}$ can be denoted by $\mathbf{H} \in \mathbb{R}^{|\mathcal{V}| \times |\mathcal{E}|}$. The degree of node is $d(v) = \sum_{e \in \mathcal{E}} w(e)h(v, e)$ and the degree of edge $\delta(e) = \sum_{v \in \mathcal{V}} h(v, e)$. $\boldsymbol{D}_e$ and $\boldsymbol{D}_v$ denotes the matrices of edge degrees and node degrees. The spectral convolution of $\mathbf{x}$ and filter $\mathbf{g}$ can be formulated as $\mathbf{g} \star \mathbf{x} = \Phi((\Phi^T \mathbf{g}) \odot (\Phi^T \mathbf{x})) = \Phi g(\Lambda)\Phi^T \mathbf{x}$, where $\odot$ denotes the element-wise multiplication and $g(\Lambda)$ is a function of Fourier coefficients Feng et al. (2019b). As simplified in GCN, the convolution operation can be simplified to $\mathbf{g} \star \mathbf{x} \approx \theta \mathbf{D}_v^{-1/2} \mathbf{HWD}_e^{-1} \mathbf{H}^T \mathbf{D}_v^{-1/2} \mathbf{x}$.

# 3 DYNAMIC HYPERGRAPH CONVOLUTIONAL NETWORKS

## 3.1 FORMULATION OF DYNAMIC HYPERGRAPH

Dynamic hypergraph can be formulated into two categories: discrete-time and continuous-time dynamic hypergraph. The discrete-time approach views dynamic hypergraph as a collection of static graph snapshots over time, while the continuous counterpart extracts fine-grained temporal information on nodes and hyperedges which characterize the dynamic evolution of hypergraph.

**Discrete-time Dynamic Hypergraph**  Discrete-time dynamic hypergraph can be formulated as $\mathcal{G}^D = (\mathcal{V}^t, \mathcal{E}^t, \mathbf{A}^t, \mathbf{H}^t, \mathbf{X}^t)$, where $\mathbf{X}^t = [\boldsymbol{x}_1^t, \boldsymbol{x}_2^t, \cdots, \boldsymbol{x}_n^t]^T \in \mathbb{R}^{n \times d}$, $\mathbf{H}^t = [\boldsymbol{h}_1^t, \boldsymbol{h}_2^t, \cdots, \boldsymbol{h}_m^t]^T \in \mathbb{R}^{m \times c}$, $\boldsymbol{x}_i^t(i = 1, 2, \cdots, n)$ denotes the feature of the $i$-th node and $\boldsymbol{h}_j^t(j = 1, 2, \cdots, m)$ denotes the feature of the $j$-th hyperedge, and $m$, $n$ is the number of hyperedges and nodes on hypergraph $\mathcal{G}^t$ (hypergraph on time step $t$). $\mathbf{A}^t \in \mathbb{R}^{n \times m}$ is an incidence matrix with binary value indicating the connectedness of nodes on hypergraph $\mathcal{G}^t$. $\mathcal{V}^t$ is the set of nodes, $\mathcal{E}^t$ is the set of hyperedges. $\boldsymbol{C}_e^t = [\boldsymbol{u}_1^t, \boldsymbol{u}_2^t, \cdots, \boldsymbol{u}_{k_e^t}^t]^T \in \mathbb{R}^{k_e^t \times d}$ and $\boldsymbol{D}_u^t = [\boldsymbol{e}_1^t, \boldsymbol{e}_2^t, \cdots, \boldsymbol{e}_{k_u^t}^t]^T \in \mathbb{R}^{k_u^t \times c}$ are used to denote the node set contained in hyperedge $e$ and the hyperedge set containing the node $u$ at time setp $t$ respectively. Note that $k_e^t$ and $k_u^t$ are the number of nodes in hyperedge $e$ and the number of hyperedges containing node $u$ on time $t$, respectively. As the representation evolve over time, we capture the spatial dependency by hypergraph convolutional networks and use CNNs to model the temporal dependency.

**Continuous-time Dynamic Hypergraph**  Continuous-time dynamic hypergraph can be defined as $\mathcal{G}^C = (\mathcal{G}^0, \mathcal{U})$ where $\mathcal{G}^0$ is the initial status (a hypergraph) and $\mathcal{U} = \{(p^t, v^t, a^t)|t <= T\}$ is a streaming of updates. $p^t$ denotes the target variable (*e.g.,* a row of $\mathbf{X}$) changed at time $t$; $v^t$ denotes the latest value of the target variable, $a^t$ denotes the action of change, including add, delete, update. Due to a static hypergraph model can be extended to dynamic hypergraphs by applying it on each snapshots and then aggregating the results of the model, and the distinction between an evolving and a temporal network is less important Skarding et al. (2020), we adapt discrete-time dynamic hypergraph to build the DyHCN model in our work.

**DyHCN**  DyHCN is composed of two modules: hypergraph convolution (HC) and temporal evolution (TE). The HC module is designed to aggregate features among nodes and hyperedges with attention mechanisms and update the embeddings of centroid nodes. The TE module is used for capturing dynamic changes in temporal features. The framework of DyHCN is illustrated in Fig.2,

## 3.2 HYPERGRAPH CONVOLUTION

Hypergraph convolution consists of three submodules: inner attention, outer attention, and embeddings update. In particular, inner attention aggregates nodes' features to hyperedge, outer attention uses attention mechanisms to determine the importance of each hyperedge, and embeddings update submodule aggregates node-hyperedge features, hyperedge features and the node features to update centroid node embeddings with the weight of each hyperedge.

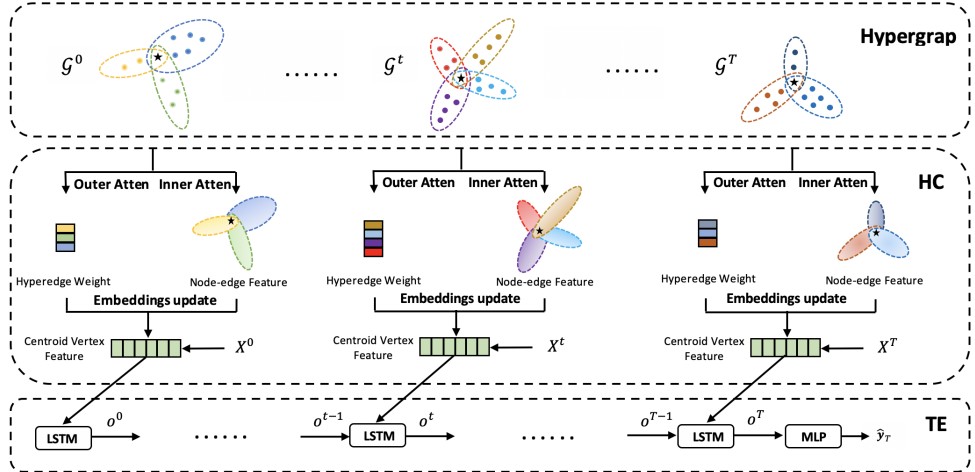

Figure 2: The framework of DyHCN: HC module consists of inner attention, outer attention and embeddings update submodules, which aggregates various features to centroid vertex, and TE module extracts temporal features for prediction.

**Inner attention**   The inner attention is shown on the left plane of Fig. 3 which aggregates node embeddings to node-hyperedge features by using a self-attention mechanism. With a multi-layer perceptron (MLP) we can get the weight score of each node. For a specific node $\boldsymbol{x}_i^t$ on time step $t$, the input of inner attention is $\boldsymbol{C}_e^t = [\boldsymbol{u}_1^t, \boldsymbol{u}_2^t, \cdots, \boldsymbol{u}_{k_e^t}^t]^T \in \mathbb{R}^{k_e^t \times d}$ and the output of node-hyperedge embedding $\boldsymbol{d}^t$ is the weighted sum of node features, which is formulated as:

$$\boldsymbol{\omega}^t = softmax(\boldsymbol{C}_e^t \mathbf{w}_e + \mathbf{b}_e), \tag{1}$$

$$\boldsymbol{d}^t = \sum_{j=0}^{k_e^t} \omega_j^t \boldsymbol{u}_j^t, \tag{2}$$

where $\mathbf{w}_e \in \mathbb{R}^{d \times 1}$ and $\mathbf{b}_e \in \mathbb{R}^{k_e^t \times 1}$ are trainable parameters, $\boldsymbol{\omega}^t \in \mathbb{R}^{k_e^t \times 1}$ is the weight of nodes in hyperedge, $\boldsymbol{d}^t \in \mathbb{R}^{1 \times d}$ denotes the node-hyperedge features, and $k_e^t$ denotes the number of nodes in hyperedge, $d$ is node feature dimension.

**Outer attention**   Due to multiple hyperedges related to center node, and the importance of each hyperedge is different, we propose an outer attention submodule to determine the weight of each hyperedge. The right plane of Fig. 3 shows the outer attention submodule which calculates the weight of each hyperedge based on hyperedge features. For specific node $\boldsymbol{x}_i^t$, the input of outer attention is $\boldsymbol{D}_u^t = [\boldsymbol{e}_1^t, \boldsymbol{e}_2^t, \cdots, \boldsymbol{e}_{k_u^t}^t]^T \in \mathbb{R}^{k_u^t \times c}$, a hyperedge set containing vertex $\boldsymbol{x}_i^t$, and the output is $\boldsymbol{\omega}_h^t$, the weight of each hyperedge on time step $t$.

$$\boldsymbol{r}_u^t = sigmoid(\boldsymbol{D}_u^t \mathbf{w}_u + \mathbf{b}_u), \tag{3}$$

$$\boldsymbol{\omega}_h^t = softmax(\boldsymbol{r}_{\boldsymbol{u}}^t), \tag{4}$$

where $\mathbf{w}_u \in \mathbb{R}^{c \times 1}$, $\mathbf{b}_u \in \mathbb{R}^{k_u^t \times 1}$ are trainable parameters and $\boldsymbol{\omega}_h^t \in \mathbb{R}^{k_u^t \times 1}$ is the weight vector of each hyperedge, $k_u^t$ is the number of hyperedges containing vertex $\boldsymbol{u}$ at time step $t$, and $c$ is the hyperedge feature dimension.

**Embeddings Update**   With the output of inner attention and out attention, we update the centroid node embeddings $\boldsymbol{s}_i^t$ by aggregating node's input features $\boldsymbol{x}_i^t$, node-hyperedge features $\boldsymbol{d}^t$ and hyperedge features $\boldsymbol{h}_i^t$ with the weight of hyperedges $\boldsymbol{\omega}_h^t$. We explore three aggregation methods, **1) Concatenated features** We concatenate the node-hyperedge features and hyperedge features directly with the activation funciton of $tanh$, $\boldsymbol{q}^t = tanh[\boldsymbol{d}^t : \boldsymbol{h}_i^t] \in \mathbb{R}^{1 \times (d+c)}$.
**2) Dot-product features** We multiply the node-hyperedge features with hyperedge features with the element-wise operation to model the interaction of two kinds features, $tanh$, $\boldsymbol{q}^t = tanh[\boldsymbol{d}^t \odot \boldsymbol{h}_i^t] \in$

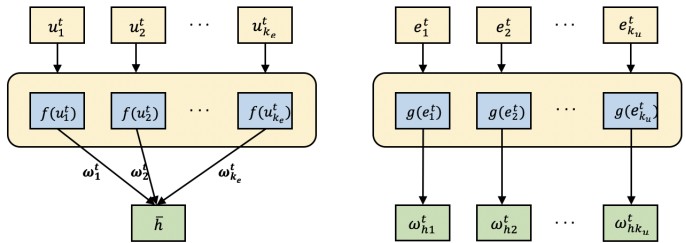

Figure 3: Inner attention on the left and outer attention on the right.

$\mathbb{R}^{1 \times d}$ (by setting $d=c$), where $\odot$ denotes element-wise product operation. **3) MLP features** We concatenate the node-hyperedge features with hyperedge features with an MLP process to aggregate the features, $q^t = tanh([d^t : h_i^t]\mathbf{W}_c + \mathbf{b}_c) \in \mathbb{R}^{1 \times d}$, where $\mathbf{W}_c \in \mathbb{R}^{(d+c) \times d}$, $\mathbf{b}_c \in \mathbb{R}^{1 \times d}$ are trainable parameters. Note that, $h_c^t$ only stands for the concatenated features for one hyperedge, so for $k_u^t$ hyperedges, we can get a concatenated features matrix $\mathbf{Q}_i^t = [q_0^t, q_1^t, \cdots, q_{k_u^t}^t]^T$ which denotes the influence from nodes and each hyperedge.

Considering the weight of each hyperedge $\omega_h^t$, we first calculate the weighted sum of concatenated features $\mathbf{Q}_i^t$ to measure the influence from all hyperedges and related nodes. And then update the specific node embedding $s_i^t$ with the input feature $x_i^t$ and the influence embeddings.

$$z_i^t = sum(\omega_h^t \cdot \mathbf{Q}_i^t), \tag{5}$$

$$s_i^t = tanh([x_i^t : z_i^t]\mathbf{W}_h + \mathbf{b}_h), \tag{6}$$

where $z_i^t \in \mathbb{R}^{1 \times d}$ is the weighted aggregated features, $\mathbf{W}_h \in \mathbb{R}^{2d \times d}$ and $\mathbf{b}_h \in \mathbb{R}^{1 \times d}$ are trainable parameters.

### 3.3 TEMPORAL EVOLUTION

The centroid node embeddings extracted by HC are independent on different time steps, we will get embeddings for each centroid node $i$ along with time, *i.e.*, $\mathbf{S}_i = [s_i^0, s_i^1, \cdots, s_i^t]^T$. We adopt a temporal evolution module to process temporal information extraction. The TE module utilize the LSTM model to extract temporal features which can be used for classification or regression tasks.

$$\mathbf{O}_i = LSTM(\mathbf{S}_i), \tag{7}$$

$$\hat{\mathbf{y}}_i = (tanh(\mathbf{O}_i\mathbf{W}_o + \mathbf{b}_o))\mathbf{W}_y + \mathbf{b}_y, \tag{8}$$

where $\mathbf{O}_i \in \mathbb{R}^{1 \times dim}$ is the temporal features extracted by LSTM, $dim$ is the hidden dimension of LSTM. $\mathbf{W}_o \in \mathbb{R}^{dim \times k}$, $\mathbf{b}_o \in \mathbb{R}^{1 \times k}$, $\mathbf{W}_y \in \mathbb{R}^{k \times l}$, $\mathbf{b}_y \in \mathbb{R}^{1 \times l}$ are trainable parameters, $k$ is the hidden dimension size for MLP and $l$ is the final output size which determined by detail task.

## 4 RELATED WORK

**GCN on regular graphs** Existing graph-based learning solutions are divided into two directions: spectral methods and spatial methods. Spectral graph convolution transform features to the spectral domain by using graph Laplacian eigenvectors and then conclude node or graph features by spectral convolution. However, the computation cost in Laplacian factorization is expensive, Defferrard et al. (2016) introduced Chebyshev polynomials to approximate Laplacian eigenvectors, and further, Kipf & Welling (2016) simplified the process by a localized first-order approximation of spectral graph convolutions. On the other side, spatial methods generate node embeddings by aggregating neighborhoods' features. GraphSAGE generates embeddings by sampling and aggregating features from a node's local neighborhood Hamilton et al. (2017). GAT leverage self-attentional mechanism and weighting different nodes in a neighborhood to generate node embeddings Veličković et al. (2017). The graph-based learning limits the relationships into pairwise, however, in many applications, the relations between objects are in higher-order that cannot be formulated by a graph structure.

**GCN on hypergraph** To evaluate the higher-order relations between nodes, Zhou et al. (2007) introduced the first hypergraph learning, where multiple nodes share the same hyperedge. On the

Table 1: Details of Tiingo and Stockwits datasets

| Dataset | Nodes | Avg hyperedges | Training days | Validation days | Testing days |
|---|---|---|---|---|---|
| Tiggo | 91 | 691 | 529 | 151 | 76 |
| Stocktwits | 91 | 177.1 | 712 | 203 | 103 |

direction of spectral methods, considering different subsets of nodes in the same hyperedge may have different structural importance, Li & Milenkovic (2017) proposed inhomogeneous hypergraph partitioning model to assign different costs to different hyperedge cuts. Li & Milenkovic (2018) defined the notion of p-Laplacians which constitute the basis of new spectral hypergraph clustering methods. Chan et al. (2018) considered a stochastic diffusion process and introduces a new hypergraph Laplacian operator generalizing the Laplacian matrix of graphs. Yadati et al. (2019) simplified hypergedges into simple edges with mediators and demonstrate the effectiveness through detailed experiments. The other way, spatial methods, Gilmer et al. (2017) raised a Message Passing Neural Networks (MPNN) framework which learns a message-passing algorithm and aggregate features for node representation. Feng et al. (2019b) introduced the first hypergraph deep learning method hypergraph neural network (HGNN). However, most of the existing works focus on static hypergraph structure which has little effort on optimizing the hypergraph structure during the learning process. DHSL Zhang et al. (2018) is the first dynamic hypergraph structure learning method that optimizes the label projection matrix and the hypergraph structure itself simultaneously. But DHSL fails to exploit high-order relations among features Jiang et al. (2019). DHGCN Jiang et al. (2019) proposed a stacked layers framework to evaluate the dynamic hypergraph with KNN to build the dynamic hyperedges. However, the input of DHGCN is fixed, which means the relations among nodes are fixed and the hypergraph structure just update on each layer. But in the real world, the relations of nodes may be connected temporarily, and existing models cannot process temporary connections or change connections among different nodes.

## 5 EXPERIMENTS

The DyHCN model can be applied to various tasks that can be formulated as dynamic hypergraph. In our work, we adopt DyHCN with news and social comment datasets for stock price prediction.

### 5.1 EXPERIMENTAL SETTING

**Tiingo dataset** [1]. The dataset covers the news content, stocks contained in the news, and the release time of news. On a specific trading day, there are varity news and each news many contains different numbers of stocks, so we construct a hypergraph with news as hyperedges and stocks as nodes. We construct the dynamic hypergraph based on crawled news from June 22, 2016 to June 23, 2019, a total of 756 trading days with one hypergraph on each trading day. Inspired by Chen et al. (2019), we adapt Finance Event Dictionary (TFED) to extract fine-grained events, and pick out the most activate 91 stocks on market for price prediction.

**Stockwits dataset** The Stocktwits dataset is a stock comment dataset which can be crawled from the web of stockwits [2]. The dataset covers the stock comment content, stocks mentioned in the comment, and the comment time. On a specific trading day, we construct a hypergraph with comments as hyperedges and stocks as nodes. We pick out 91 stocks with the highest market value in different industries on S&P 500 for price prediction, and collect the data from Aug. 7, 2014 to Aug. 20, 2018, a total of 1015 trading days with one hypergraph on each trading day. The details of datasets are shown on Table 1.

With the construction of dynamic hypergraph, we assign the nodes featuress with the hidden embedding of price and volume extracted by LSTM, and hyperedges featrues with the embedding represented by GloVe Pennington et al. (2014). The feature dimension of hyperedges and nodes are set to 50. The training, validation, and testing sets are separated as in Table 1. To measure the result

---

[1]https://www.tiingo.com
[2]https://stocktwits.com

Table 2: Performance on Tiingo and Stocktwits dataset with hidden size 128 of TE

| Dataset | Tiingo | | | | Stocktwits | | | |
|---|---|---|---|---|---|---|---|---|
| Models | MAE | MAPE | MSE | Promote | MAE | MAPE | MSE | Promote |
| DA-RNN | 0.1869 | 0.4786 | 0.0467 | 55.37% | 0.1855 | 1.0950 | 0.0384 | 71.55% |
| HAN | 0.1466 | 0.3017 | 0.0548 | 42.43% | 0.1272 | 0.5517 | 0.0414 | 61.29% |
| RSR | 0.0946 | 0.2596 | 0.0183 | 7.57% | 0.0893 | 0.1870 | 0.0158 | 18.19% |
| DHGCN | 0.0984 | 0.2754 | 0.0216 | 15.08% | 0.0845 | 0.1782 | 0.0156 | 14.86% |
| DyHCN | **0.0873** | **0.2533** | **0.0160** | | **0.0732** | **0.1660** | **0.0118** | |

of different models for prediction, we use three evaluation metrics, the mean squared error (MSE), mean absolute error (MAE) and mean absolute percentage error (MAPE).

**Baselines** To evaluate the result of our proposed DyHCN model, we compare the experiment result with the traditional time series, NLP-based, graph-based and hypergraph-based model: **1) DA-RNN** Hsu et al. (2009) One of the state-of-the-art models for time series prediction. **2) HAN** Hu et al. (2018) The representations of the NLP model for stock price prediction. **3) RSR** Feng et al. (2019a) The state-of-the-art graph-based model for price prediction. **4) DHGCN** Jiang et al. (2019) The hypergraph-based model for prediction. Because the model RSR and DHGCN are designed for static graph/hypergraph, we present the RSR and DHGCN for daily price prediction. To compare with baseline models, we use DyHCN with stacked layer HC and TE module for price prediction, and add a dropout layer with the dropout rate of 0.5 before TE module. We set the learning rate 0.005 and training epoch 1000.

## 5.2 RESULTS AND ANALYSIS

We report the performance of all methods in Table 2, Table 3 and Fig. 4. From Table 2, we have the following observations:

**1)** Compared with DA-RNN, the MAE and MAPE scores of HAN decreases by 21.56% and 36.96% from 0.1869, 0.4786 to 0.1466, 0.3017, while the MSE increases by 17.34% from 0.0467 to 0.0548, indicating that the extra features such as Tiingo or Stocktwits comment data are useful for stock price prediction.

**2)** The MAE, MAPE and MSE results of RSR outperform HAN, and decrease by 35.47%, 13.95% and 66.61% respectively, indicating that the consideration of relations between different stocks would improve the performance of the prediction result.

**3)** Comparing the graph-based model RSR with hypergraph-based model DHGCN, there is no significant difference in the performance of the model. However, in the Tiingo dataset, the results of RSR decrease by 3.86%, 5.74%, 15.28% compared with DHGCN respectively, while in the Stocktwits comment dataset, the model shows the opposite result, the three metrics of RSR increase by 5.68%, 4.94%, 1.28% comparied with DHGCN. This shows that the performance of RSR and DHGCN models are not stable on different datasets.

**4)** The results of DyHCN outperform RSR and DHGCN. On Tiingo dataset, the MAE decreases by 7.72%, 11.28%, the MAPE decreases by 2.43%, 8.02%, and the MSE decreases 12.57%, 25.93% compared with RSR and DHGCN respectively. On Stocktwits comment dataset, the MAE decreases by 18.03%, 13.37%, the MAPE decreases by 11.23%, 6.85%, and the MSE decreases by 25.32%, 24.36% compared with RSR and DHGCN respectively. The result shows that in both Tiingo and Stocktwits comment datasets, the performance of DyHCN would keep stable, and with the consideration of dynamic information, the performance is better than static graph/hypergraph based model.

**5)** Comparing DyHCN with DA-RNN, HAN, RSR and DHGCN, the average loss of MAE, MAPE and MSE decrease by 55.37%, 42.43%, 7.57% and 15.08% on Tiingo dataset respectively, and 71.55%, 61.29%, 18.19%, 14.86% on Stocktwits comment dataset respectively.

To test the stability and the scalability of the model, we evaluate different feature aggregation methods. Fig. 4 shows the performance of different feature aggregation methods with a hidden size of 16, 32, 64, and 128 on TE module. Comparing the results, the MLP feature concatenate method reminds stable and outperforms the cat and multi-feature aggregate methods. Besides, with the comparison

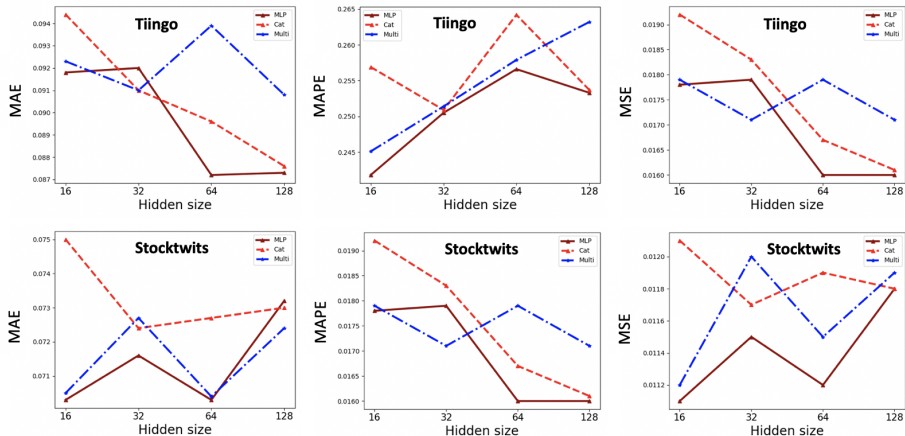

Figure 4: Performance of different features aggregate methods on Tiingo and Stocktwits social comment datasets with the hidden size of 16, 32, 64 and 128 of TE.

Table 3: Performance on Tiingo and Stocktwits dataset with hidden size 128 of TE

| Dataset | Tiingo | | | Stocktwits | | |
|---|---|---|---|---|---|---|
| Models | MAE | MAPE | MSE | MAE | MAPE | MSE |
| DyHCN(no inner) | 0.1303 | 0.3988 | 0.0347 | 0.0745 | 0.1667 | 0.0122 |
| DyHCN(no outer) | 0.0887 | 0.2613 | 0.0165 | 0.0732 | 0.1695 | 0.0120 |
| DyHCN(no TE) | 0.0982 | 0.2754 | 0.0218 | 0.0842 | 0.1784 | 0.0160 |
| HGCN(with TE)Feng et al. (2019b) | 0.2943 | 0.8025 | 0.1771 | 0.2943 | 0.8025 | 0.1771 |
| DyHCN | **0.0873** | **0.2533** | **0.0160** | **0.0732** | **0.1660** | **0.0118** |

of results with different hidden size, the performance has no significant difference, indicating that the hidden size of LSTM is not the major factor for model prediction.

In addition to the comparison with baselines above, we also evaluate the effectiveness of submodules including inner attention, outer atteniton, HC and TE. We evaluate the effectiveness of each module which shown in Table 3, we use DyHCN without inner attention, DyHCN without outer attention and DyHCN without TE to evaluate the effectiveness of inner, outer attention and TE module. Also, we use HGCN model Feng et al. (2019b) which aggregate node features on static hypergrap to replace HC module to evaluate the effectiveness of HC module. The result shows that, on both datasets, the performance of DyHCN are better than DyHCN without inner, outer and TE module. The inner and outer attention module determines the importance of each message, and passes the corresponding information to the centroid which would be used for prediction more acurrate. The TE module considers the impact of previous information and extracts the temporal features from series embeddings, which would be better than from individual time step. In addition, the performance of HGCN Feng et al. (2019b) with TE model is also worse than DyHCN, while HGCN works well on static hypergraph, indicating that DyHCN is more suitble for dynamic hypergraph tasks.

## 6 CONCLUSION

In this paper, we proposed a framework of dynamic hypergraph convolutional networks (DyHCN), which consists of hypergraph convolution (HC) and temporal evolution (TE) module. The HC is delicately designed by equipping inner attention and outer at-tention, which adaptively aggregate nodes' features to hyperedge and estimate theimportance of each hyperedge connected to the centroid node, respectively. And then update the centroid node embeddings by aggregating various related features. The TE captures the long-range temporal information of dynamic hypergraph features. Based on the two modules, the DyHCN get the dynamic relations between different nodes with the dynamic weight of hyperedges on different time steps. DyHCN can be used for various tasks that can be formulated as dynamic hypergraph, and extensive experiments on the newly constructed Tiingo and Stocktwits comment datasets show that our proposed DyHCN outperforms the state-of-the-art model for the modeling of dynamic hyper-order relations.

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
