# OpenReview forum: "DyHCN: Dynamic Hypergraph Convolutional Networks"
_ICLR.cc/2021/Conference — Reject_

### Official Review · AnonReviewer2 · 2020-10-29
**Proposal for a dynamic hypergraph neural network**

**Rating:** 4
**Confidence:** 4

**Review:**

The paper extends over hypergraph convolutional networks (HCN) by adding a temporal evolution module in order to solve prediction tasks in a dynamic environment. The main part of the paper is the description of the proposed system. It is composed of a HCN for computing node embeddings at each time step and a LSTM as the temporal module. Experimental results are provided for dynamic prediction tasks over stock datasets.

The proposed system seems to be a reasonable choice for solving prediction tasks on dynamic hypergraphs. But, in my opinion, some choices could have been justified and variations of the system could have been compared. The only considered system variation is the aggregation method and the best one is the only one using trainable parameters. Thus I think that there is room for improvement. Also the current version is difficult to read due to many typos, mispellings, repetitions, ... Therefore, in my opinion, the paper is not ready for publication.

Detailed comments.

* Too many typos, mispellings, ... to be given in this review

* Introduction. It could be improved. Formal definitions are not necessary. I am not convinced that the two formulations for the evolution are useful and they are repeated in Section 3.1. The presentation of the system could be improved: the description of the "two challenging toughs" is not convincing enough; the description of the DyHCN system is unclear (for instance what is a node-hyperedge ?).

* Section 2. §GCN. There is a huge recent literature on GNNs. Please give useful links and please explain why you choose to only present some GCNs. The definition is not easy to read.

* Section 2. §HCN. Also many HNNs models have been proposed. Please justify your choice. The definition is not easy to read. Also definitions of hypergraphs should be coherent (definition in Section 2 wrt definition in section 3.1).

* Section 3.1. It is already presented in the introduction. I am not convinced that the continuous-time presentation is useful because it is not used.

* Section 3. Please introduce what is the meaning of a "centroid node"

* Section 4. §GCN should be updated with recent references.

* Section 5.1. I would like to know why it is useful to model the task with hypergraphs. Why is it important to know for a stock event in which news or which comment it appears?

* Section 5. It is not clear to me how the authors model the task when using time series prediction. Also for RSR and DHGCN.

---

> ### Author Response · Authors · 2020-11-16
> **We rewritten parts of the manuscript and corrected the spelling and grammatical errors in the manuscript.**
>
> Thank you for your insightful comments.
>
> Following your comments, we have updated the manuscript with more recent references, more explanations on the experiment.
>
> Q1. The introduction could be improved. The description of the "two challenging toughs" is not convincing enough; the description of the DyHCN system is unclear
>
> A1. In the introduction, we introduced the discrete-time and continuous models as two feasible solutions to dynamic hypergraph and defined the formulation to compare the difference between the two models. Due to a static hypergraph model can be extended to dynamic hypergraphs by applying it on each snapshot and then aggregating the results of the model, and the distinction between an evolving and a temporal network is less important, so we adapt discrete-time formulation in our work. Also, in section 3.1, we deleted the expression about continuous-time models, and focus on the discrete-time models throughout the manuscript.
>
> In terms of the two challenging toughs, the question we considered is dynamic hypergraph, the hypergraph various over time, which means the neighborhoods of specific node changes along time. For example, in the price prediction task, we can construct a hypergraph based on news on each trading day, and each hyperedge is constructed with the nodes contained in corresponding news. While, the number of nodes various from different news, and the number of news also vary from each day. So, in our work, the first challenging is how to design a proper solution to extract node embeddings from the various nodes in hyperedge and the different numbers of hyperedges connected to the node. Besides, the node embeddings extracted from the hypergraph are the representations on each snapshot, while with the new message comes, new hypergraph generated, and the node embeddings also updated with the newly hypergraph. So, the node embeddings on each snapshot are related to previous information, and the second challenge in our work is how to extract the temporal dependency features into node embeddings for price prediction.
>
> We have modified the description of the DyHCN, and explained the ambiguity of the model, for instance, the definition of node-hyperedge, ‘Firstly, inner attention transform node features in hyperedge into the node-hyperedge feature which indicates the aggregated features from node to hyperedge’ etc., so that the manuscript can be read more clearly.
>
> Q2. Section 2. the definition of GCN
>
> A2. We updated the formulation and expression of GCN in section 2. We introduced two typical models of GCN in the spatial domain. In the following research, most models are improved based on these two models, so we introduced these two models in the Preliminary section.
>
> Q3. Section 2. the definition of HCN
>
> A3.  We updated the formulation and expression of HCN in section 2. In terms of model selection, we introduced the latest two solutions that are widely used on hypergraphs. The definition of a hypergraph in section 2 is the preliminary of a static hypergraph, and in section 3.1 the definition is about discrete-time dynamic hypergraph which is the problem we focus on in this manuscript.
>
> Q4. Section 3.1. It is already presented in the introduction
>
> A4.  We modified the part of the introduction and analyzed the pros and cons, and decided to focus on the discrete-time dynamic hypergraph in our work, so we delete this part of the statement.
>
> Continued on the next section.

---

> > ### Author Response · Authors · 2020-11-23
> > **Continue the reply in the previous section.**
> >
> > Q5 Section 3., the meaning of a "centroid node"
> >
> > A5. Considering that each node has a different number of neighbor nodes, we use the centroid node to represent the nodes that need to aggregate neighbor features. To avoid ambiguity, we change the description of “centroid node” to “specific node”, and modified the description to ‘Taking each node with a different number of neighbors as a specific node, the HC module aggregates features among...’.
> >
> > Q6. Section 4. §GCN should be updated with recent references
> >
> > A6. We have updated 10 related references from 2018 to the present and gave a brief introduction.
> >
> > Q7. Section 5.1. The reason for use hypergraph and experiment settings
> >
> > A7. We have rewritten section 5.1 and explained why dynamic hypergraph is effective in solving our task. Stock prices are affected by many factors, in addition to the macro factors, the related news/comment of listed companies in the stock market will also affect stock prices. While one news or comment may contain different numbers of stocks, which means not only the news or comment itself have an inﬂuence on these stocks, but also these stocks affect each other. The experiments are designed to show how the proposed model can effectively use dynamic hypergraphs to model the inﬂuence of both the stock news/comments itself and the interaction of stocks contained in news/comments for stock price prediction.
> >
> > In the modified section 5.1, we introduced Objectives, Dataset description, Experiment settings, and Baselines. In objectives, we explained why dynamic hypergraph is effective in solving our task. In the section of dataset description, we introduced the details of the dataset, including the dataset fields, dataset split, and the input, output of our proposed model, besides, we introduced the hypergraph construction method in the appendix. In experiment settings, we explained the settings of our proposed model, including the initial input, normalize method and the settings of hyperparameter. In the baselines section, we added the HGC-RNN model as a baseline which models time-series hypergraph for prediction and introduced the hyperparameter settings of five baselines in detail. Additionally, we added an NYC-Taxi dataset for taxi demand prediction to show our model is effective for time series prediction.
> >
> > Additionally, we checked the spelling and grammar of the manuscript many times to avoid errors.

---

### Official Review · AnonReviewer3 · 2020-10-29
**A decent formulation for discrete-time dynamic hypergraphs with some improvement in prediction accuracy**

**Rating:** 6
**Confidence:** 4

**Review:**

This paper proposes a method called DyHCN for learning dynamic hypergraph convolutional networks where the hypergraph structure is allowed to evolve over time. The interactions within each hyper edge, that between nodes, as well as related are used to learn the hypergaph embedding. The evolution of the centroid nodes is then modelled using LSTM. DyHCN gives better modelling accuracy as compared to some existing ones.

Pros:
- The formulation is clearly presented.
- The inner and outer attention models adopted are reasonable
- Empirical experiments have been carried out to verify its effectiveness.

Cons:
- Only discrete-time dynamic hypergraph is considered.
- The attention model and the modelling of the evolution of the centroid nodes are not particularly novel.
- The performance improvement as compared with the SOTA method is incremental.
- Only one particular prediction is adopted for the performance evaluation.

Specific comments
- In Inner attention section, x_i^t is referred to as a nod but it should be th feature of node I

Qn:
- Other than stock price prediction, what are the other possible prediction tasks? Can the performance comparison be carried out based on the additional predication tasks?

---

> ### Author Response · Authors · 2020-11-16
> **Added the NYC data set for taxi demand forecasting, and revised part of the manuscript statements to make it clearer.**
>
> Thank you for your insightful comments.
>
> Following your comments, we have updated the manuscript with more recent references, more explanations on the experiment, and added an NYC-Taxi dataset for the taxi demand prediction task.
>
> Q1. An only discrete-time dynamic hypergraph is considered.
>
> A1. In the introduction, we introduced the discrete-time and continuous models as two feasible solutions to dynamic hypergraph and defined the formulation to compare the difference between the two models. Due to a static hypergraph model can be extended to dynamic hypergraphs by applying it on each snapshot and then aggregating the results of the model, and the distinction between an evolving and a temporal network is less important, so we adapt discrete-time formulation in our work. Also, in section 3.1, we deleted the expression about continuous-time models, and focus on the discrete-time models throughout the manuscript.
>
> A2. The attention model and the modeling of the evolution of the centroid nodes are not particularly novel.
>
> Q2. We agree that the key idea of our work is based on the attention model, which has been used in a graph-based model for modeling feature interactions. However, we use it to model the interaction between nodes and hyperedges in the dynamic hypergraph, which is a rather different scenario and has not studied by previous work. In our opinion, appropriately adapting an existing technique to solve a different problem should also be treated as a significant contribution. In fact, there have been many inﬂuential papers that adapt/introduce an existing technique to better solve another problem. For example, the neural attention mechanism is ﬁrst developed in computer vision [1] and then adapted to machine translation by [2] and to GNN by [3]. Such adaption of existing technique does not aﬀect the contribution signiﬁcance of [2] and [3], which later are proved to be inﬂuential work.
>
> [1] Mnih et al. Recurrent models of visual attention. NIPS 2014
>
> [2] Bahdanau et al. Neural machine translation by jointly learning to align and translate. ICLR 2015
>
> [3] Velicˇkovic ́ et al. Graph Attention Network. ICLR 2018.
>
> Q3. In the Inner attention section, x_i^t is referred to as a nod but it should be th feature of node I.
>
> A3. In the inner attention section, $x_i^t$ is the feature of node I, so we charge the description to ‘For a specific node $i$ on the time step $t$, the input of inner attention is …’.
>
> Q4. Other than stock price prediction, what are the other possible prediction tasks? Can the performance comparison be carried out based on the additional prediction tasks?
>
> A4. We add an NYC-Taxi dataset for taxi demand prediction, and the result shows that our model outperforms other baselines.
>
>     Models   MAE      MAPE     MSE     Promote
>
>     RSR     0.0182   0.0907   0.0061   10.51%
>     DHGCN   0.0186   0.0936   0.0059   11.31%
>     HGC-RNN 0.0220   0.1032   0.0092   28.65%
>
>     DyHCN   0.0175   0.0864   0.0047
>
> Additionally, we checked the spelling and grammar of the manuscript many times to avoid errors.

---

### Official Review · AnonReviewer4 · 2020-10-31
**Dynamic hypergraph convolutional networks applied to financial data**

**Rating:** 6
**Confidence:** 3

**Review:**

This paper proposed a pipeline for dynamic hypergraph convolutional networks, with a two-fold component that handles both the hypergraph convolution and the temporal evolution. I think the paper is a nice contribution to the less-well studied area of time dependent networks, and to some extent, also that of hypergraph embedding. The application to finance and the data sets used are interesting, with hypergraphs/edges encoding a proxy for co-occurence behaviour of stocks.

I think the experimental section is less strong, with only two data sets considered (albeit indeed the setting is less usual and less such data sets exist), and the authors only compared against a single other methodology. Regarding the performance metrics, the 3 ones currently used are less effective/appropriate when dealing with financial/stock price prediction data, where more relevant measures would be PnL and Sharpe Ratio, especially when weights are taken into account that relate to the liquidity of each of the instruments considered in the portfolio. The number of testing days is also very small, in order to be able to draw meaningful results, especially for a portfolio of 91 instruments.

The paper is for the most part fairly straightforward to follow, though it is full of typos: a few examples being
an financial, this work explore, an node, Spectral graph convolution transform features, outer at-tention, to name a few.
The paper should be proofread very carefully.

Some of the notation could be better explained. I don’t see any mention of what y represents? (future price/returns)

---

> ### Author Response · Authors · 2020-11-16
> **Added the NYC data set for taxi demand forecasting, and revised part of the manuscript statements to make it clearer.**
>
> Thank you for your insightful comments.
>
> Following your comments, we have updated the manuscript with more recent references, more explanations on the experiment, and added an NYC-Taxi dataset for the taxi demand prediction task.
>
> Our replies to the questions are listed below.
>
> Q1. The experimental section is less strong, with only two data sets considered.
>
> A1. We add an NYC-Taxi dataset for taxi demand prediction, and the result shows that our model outperforms other baselines.
>
>
>
>     Models   MAE      MAPE     MSE     Promote
>
>     RSR     0.0182   0.0907   0.0061   10.51%
>     DHGCN   0.0186   0.0936   0.0059   11.31%
>     HGC-RNN 0.0220   0.1032   0.0092   28.65%
>
>     DyHCN   0.0175   0.0864   0.0047
>
>
>
> Q2. The number of testing days is also very small, to be able to draw meaningful results, especially for a portfolio of 91 instruments.
>
> A2. We split the dataset into training, validation, and testing set according to the ratio of 7:2:1. Comparing DyHCN with other baselines under the same dataset split, and DyHCN works best, indicating that DyHCN is more suitable to solve the dynamic hypergraph problem. The length of the testing set is not the most important factor in evaluating the effectiveness of the model. Besides, we need to construct a hypergraph on each trading day and calculate the static embedding of nodes on each hypergraph, the training time will be greatly increased if a large number of data set is added.
>
> Q3. Performance metrics problem.
>
> A3. We backtested the experimental results, the initial capital is set to 10000, and the transaction fee and slippage are set to 0.008, which is an appropriate threshold. When the predicted price on the second day is higher than the current price, we buy the stock, and vice versa. We report the PnL and Sharpe for Tiingo and Stocktwits datasets, and the result shows that in Stocktwits, the annual PnL reached 43.89% and Sharpe reached 5.53. In Tiingo, the annual PnL reached 33.4% and Sharpe reached 3.87. However, our model is just a prototype for stock market strategies, and in real trading, we need to consider more details to make the strategy better.
>
> Q4. Some of the notation could be better explained.
>
> A4. We modified the description of experiment settings at Section 5.1. In the price prediction task, the input is the time series normalized open, high, low, close price and volume data denoted
> as $p_o^{0:T},p_h^{0:T},p_l^{0:T},p_c^{0:T},p_v^{0:T}$ of total $T$ trading days, and output $\textbf{Y}^{t+1}$ is the close price of the $T+1$th day.
> For a specific trading day $i$, the hypergraph $\boldsymbol{\mathcal{G}}^i=(\boldsymbol{\mathcal{V}}^i,\boldsymbol{\mathcal{E}}^i)$ is constructed
> as mentioned above, and the initial features of nodes $\boldsymbol{\mathcal{V}}^i$ are extracted by LSTM with
> the input of normalized price and volume of past ten days,
> \eg $\boldsymbol{\mathcal{V}}^i=LSTM(p_o^{i-10:i},p_h^{i-10:i},p_l^{i-10:i},p_c^{i-10:i},p_v^{i-10:i})$, and
> we set the embedding size of LSTM to 128. Also, we set the initial features of
> $\boldsymbol{\mathcal{E}}^i=\{\boldsymbol{\mathcal{E}}^{i}_0,\cdots,\boldsymbol{\mathcal{E}}^{i}_m\}$ with
> the embeddings of news/comment on day $i$, \eg for each hyperedge $\boldsymbol{\mathcal{E}}^{i}_k(k\in{[0,\cdots,m]})$,
> the embeddings are initialized by GloVe \cite{pennington-etal-2014-glove} with the $k$th news/comment, and the embedding size
> is set to 128. With the construction of
> $\boldsymbol{\mathcal{G}}=\{\boldsymbol{\mathcal{G}}^{0},\cdots,\boldsymbol{\mathcal{G}}^{i},\cdots,\boldsymbol{\mathcal{G}}^{T}\}$,
> we use the past 10 trading days normalized price, volume and hypergraphs,
> \eg $p_o^{t-10:t},p_h^{t-10:t},p_l^{t-10:t},p_c^{t-10:t},p_v^{t-10:t}$ and $\boldsymbol{\mathcal{G}}^{t-10:t}$, to predict the normalized close
> price on 11th trading day $y^{t+1}$. We set z-score normalization as the default normalization method in our experiments.
>  The model uses a stacked-layer HC with TE module for price prediction, and add a dropout layer with the dropout rate of 0.5 before the TE module. DyHCN is trained with Adam optimizer with a learning rate of 0.005,
>  the embedding size of the TE module is set to 128. To measure the results, we use three evaluation metrics, namely the mean squared error (MSE), mean absolute error (MAE), and mean absolute percentage error (MAPE).
>
> Additionally, we checked the spelling and grammar of the manuscript many times to avoid errors.

---

### Official Review · AnonReviewer1 · 2020-11-02
**Not ready enough for publication**

**Rating:** 5
**Confidence:** 3

**Review:**

Minor: Multiply typos and grammatical errors: (esp. section 5.1 onwards).
For example, see here: "With the construction of dynamic hypergraph, we assign the nodes featuress with the hidden embedding of price and volume extracted by LSTM, and hyperedges featrues with the embedding represented by GloVe Pennington et al. (2014)"

Clarity:
(1) Clear description of the problem statement and model overall. Images are well drawn
(2) Reason for sigmoid in eqn(3) is not clear
(3) How the initial node/hyperedge features are generated is not very clear . This is important given that the newly constructed Tiingo and Stocktwits in the paper are proposed as benchmarks for comparison with baseline models. And that there are no additional datasets apart from these 2.

Novelty:
(1) Similar problem statement addressed in https://dl.acm.org/doi/pdf/10.1145/3394486.3403389 - Needs to be added as a baseline.
(2) The paper proposes a static hypergraph encoding technique and feed the encoding from snapshots at different timestamps into an LSTM for time -series  prediction.
(3) The static hypergraph encoding technique in itself is not very novel. The paper uses attention mechanism for aggregation of node features into hyperedges (inner attention) and yet another attention mechanism (outer attention) for aggregating influences of hyperedges onto nodes .Finally an aggregation layer is used to update node embeddings. This setup is again similar to the paper mentioned above.

Experiments: (1) The paper proposes a mechanism for encoding hypergraphs which evolve over time. However, the authors run experiments on only 2 graphs and compared with 4 baseines.
(2) There are only two datasets both related to stock prediction. The procedure for node/hyperedge feature vector generation needs more explanation. Also, it’ll be good to run the model+baselines on additional graphs from other domains (pointer can be seen at paper linked above)

---

> ### Author Response · Authors · 2020-11-16
> **Added baseline, NYC-Taxi data set for taxi demand prediction, and revised part of the manuscript content.**
>
> Thank you for your insightful comments.
>
> Following your comments, we have updated the manuscript with more recent references, more explanations on the experiment, and added an NYC-Taxi dataset for the taxi demand prediction task.
>
> Our replies to the questions are listed below.
>
> Q1. The reason for sigmoid in Eqn(3) is not clear
>
> A1. In Eqn(3), we map the embedding of hyperedge $D_u^t$ into a one-dimensional vector, so the sigmoid activate function here is not necessary, and we change the formulation to $r_u^t=D_u^tw_u+b_u$.
>
> Q2. How the initial node/hyperedge features are generated is not very clear.
>
> A2. We modified the description of experiment settings in Section 5.1.
>
> The Tiingo and Stocktwits datasets contain similar fields, and we use the same method to construct hypergraphs.
> There are three main fields in these datasets: released time, news/comment content, and stocks contained in news/comment.
> We set news/comment as hyperedges, and the stocks contained as nodes in hyperedges. The procedure is as follows:
>
>   1.  We aggregate news/comment on a daily basis and exclude holidays.
>
>   2.  We select news/comment that contains more than 2 stocks and less than 5 stocks on each trading day. More than 2 stocks can ensure the construction of a hyperedge, while less than 5 stocks can prevent some introductory news, which will not have much influence on the price prediction.
>
>   3.  On each trading day, we use the filtered news/comment as hyperedges and the stocks in the news/comment as the nodes of the
>   corresponding hyperedges.
>
> The constructed hypergraphs is denoted as $\boldsymbol{\mathcal{G}}=\{\boldsymbol{\mathcal{G}}^{0},\cdots,\boldsymbol{\mathcal{G}}^{T}\}$ according to the order of
> trading days, where $T$ is the total trading days.
> In each hypergraph $\boldsymbol{\mathcal{G}}^i=(\boldsymbol{\mathcal{V}}^i,\boldsymbol{\mathcal{E}}^i)(i\in{[0,\cdots,T]})$,
> $\boldsymbol{\mathcal{V}}^i=\{\boldsymbol{\mathcal{V}}^i_0,\cdots,\boldsymbol{\mathcal{V}}^i_{91}\}$
> denotes the selected 91 stocks on the $i$th day.
> The hyperedges $\boldsymbol{\mathcal{E}}^i=\{\boldsymbol{\mathcal{E}}^{i}_0,\cdots,\boldsymbol{\mathcal{E}}^{i}_m\}$ are
> constructed with each news/comment on corresponding day, where $\boldsymbol{\mathcal{E}}^i_k(k\in{[0,\cdots,m]})$ denotes the $k$th news/comment
> on day $i$, and $m$ is the number of hyperedges on day $i$. We use $\boldsymbol{u}_l^i\in{\boldsymbol{\mathcal{V}}^{i}}(l\in[0,\cdots,n])$
> denote the nodes contained in the hyperedge $\boldsymbol{\mathcal{E}}^i_k(k\in{[0,\cdots,m]})$, where $n$ is the number of nodes in the hyperedge.
>
> In the price prediction task, the input is the time series normalized open, high, low, close price and volume data denoted
> as $p_o^{0:T},p_h^{0:T},p_l^{0:T},p_c^{0:T},p_v^{0:T}$ of total $T$ trading days, and output $\textbf{Y}^{t+1}$ is the close price of the $T+1$th day.
> For a specific trading day $i$, the hypergraph $\boldsymbol{\mathcal{G}}^i=(\boldsymbol{\mathcal{V}}^i,\boldsymbol{\mathcal{E}}^i)$ is constructed
> as mentioned above, and the initial features of nodes $\boldsymbol{\mathcal{V}}^i$ are extracted by LSTM with
> the input of normalized price and volume of past ten days,
> \eg $\boldsymbol{\mathcal{V}}^i=LSTM(p_o^{i-10:i},p_h^{i-10:i},p_l^{i-10:i},p_c^{i-10:i},p_v^{i-10:i})$, and
> we set the embedding size of LSTM to 128. Also, we set the initial features of
> $\boldsymbol{\mathcal{E}}^i=\{\boldsymbol{\mathcal{E}}^{i}_0,\cdots,\boldsymbol{\mathcal{E}}^{i}_m\}$ with
> the embeddings of news/comment on day $i$, \eg for each hyperedge $\boldsymbol{\mathcal{E}}^{i}_k(k\in{[0,\cdots,m]})$,
> the embeddings are initialized by GloVe \cite{pennington-etal-2014-glove} with the $k$th news/comment, and the embedding size
> is set to 128. With the construction of
> $\boldsymbol{\mathcal{G}}=\{\boldsymbol{\mathcal{G}}^{0},\cdots,\boldsymbol{\mathcal{G}}^{i},\cdots,\boldsymbol{\mathcal{G}}^{T}\}$,
> we use the past 10 trading days normalized price, volume and hypergraphs,
> \eg $p_o^{t-10:t},p_h^{t-10:t},p_l^{t-10:t},p_c^{t-10:t},p_v^{t-10:t}$ and $\boldsymbol{\mathcal{G}}^{t-10:t}$, to predict the normalized close
> price on 11th trading day $y^{t+1}$. We set z-score normalization as the default normalization method in our experiments.
>  The model use stacked-layer HC with TE module for price prediction, and adds a dropout layer with the dropout rate of 0.5 before the TE module. DyHCN is trained with Adam optimizer with a learning rate of 0.005,
>  the embedding size of the TE module is set to 128. To measure the results, we use three evaluation metrics, namely the mean squared error (MSE), mean absolute error (MAE), and mean absolute percentage error (MAPE).
>
> Continued on the next section.

---

> > ### Author Response · Authors · 2020-11-24
> > **Continue the reply in the previous section.**
> >
> > Q3. Other datasets and baseline.
> >
> > A3. We add the NYC-Taxi dataset for taxi demand prediction task, and the baseline of HGC-RNN (Hypergraph Convolutional Recurrent Neural Network) in our experiment. The result shows that our proposed model outperforms the baselines on the price prediction task and prediction task.
> >
> > There are some differences between HGC-RNN with our model. In HGC-RNN, the input is the data of previous days, such as the previous 8 days taxi demand on the NYC-Taxi dataset, and one hypergraph constructed with 8 days data. While in our model, the input is the data of previous days and 8 hypergraphs constructed with the demand on each day, which would provide richer information for prediction tasks. Additionally, please kindly note that the mentioned baseline HGC-RNN is published on 2020-09-08 which is very close to the deadline of ICLR and thus not included in our original version.
> >
> > Finally, we have checked the spelling and grammar of the manuscript many times to avoid errors.

---

### Decision · Program_Chairs · 2021-01-07
**Final Decision**

**Decision:**

Reject

**Comment:**

The paper builds upon hypergraph convolutional networks (HCN), extending them to time-varying hypergraphs in dynamical settings.  However, as some of the reviewers pointed out, it would be useful to explore other system variations to better justify the choices in this particular approach; perhaps an evaluation on a wider set of datasets would also strengthen the contribution of the paper,  as well as adding evaluation metrics that can be more appropriate for the application considered (stock market prediction). Also, concerns were raised by several reviewers regarding the somewhat incremental  improvement over the state of art, and the degree of novelty in the proposed approach. Overall, while the problem considered is important and the approach is promising, the paper in its current shape  is somewhat borderline and may require a bit of additional work to be ready for publication.